# Associations of the built environment with type 2 diabetes in Asia: a systematic review

Garudam Raveendiran Aarthi ![ORCID] ,[1,2] Thaharullah Shah Mehreen Begum,[1,3] Suzana Al Moosawi,[3] Dian Kusuma,[4] Harish Ranjani,[5] Rajendra Paradeepa ![ORCID] ,[6] Venkatasubramanian Padma,[2] Viswanathan Mohan,[6] Ranjit Mohan Anjana,[2,6] Daniela Fecht[7]

For numbered affiliations see end of article.

**Correspondence to**
Dr Daniela Fecht;
d.fecht@imperial.ac.uk

## ABSTRACT

**Objectives** Our study aimed to systematically review the literature and synthesise findings on potential associations of built environment characteristics with type 2 diabetes (T2D) in Asia.

**Design** Systematic review of the literature.

**Data sources** Online databases Medline, Embase and Global Health were used to identify peer-reviewed journal articles published from inception to 23 January 2023.

**Eligibility criteria** Eligible studies included cohort, cross-sectional and case–control studies that explored associations of built environment characteristics with T2D among adults 18 years and older in Asia.

**Data extraction and synthesis** Covidence online was used to remove duplicates and perform title, abstract and full-text screening. Data extraction was carried out by two independent reviewers using the OVID database and data were imported into MS Excel. Out of 5208 identified studies, 28 studies were included in this systematic review. Due to heterogeneity in study design, built environment and outcome definitions, a semiqualitative analysis was conducted, which synthesised results using weighted z-scores.

**Results** Five broad categories of built environment characteristics were associated with T2D in Asia. These included urban green space, walkability, food environment, availability and accessibility of services such as recreational and healthcare facilities and air pollution. We found very strong evidence of a positive association of particulate matter ($PM_{2.5}$, $PM_{10}$), nitrogen dioxide and sulfur dioxide ($p<0.001$) with T2D risk.

**Conclusion** Several built environment attributes were significantly related to T2D in Asia. When compared with Western countries, very few studies have been conducted in Asia. Further research is, therefore, warranted to establish the importance of the built environment on T2D. Such evidence is essential for public health and planning policies to (re)design neighbourhoods and help improve public health across Asian countries.

**PROSPERO registration number** CRD42020214852.

## STRENGTHS AND LIMITATIONS OF THIS STUDY

⇒ This is the first systematic review to synthesise the evidence base on association of built environmental characteristics including urban green space, walkability, food environments, availability and accessibility to services (recreational and healthcare), and air pollution with type 2 diabetes in Asia.

⇒ Due to the heterogeneity in built environment characteristics, outcome measures and methodology used in the eligible studies, a formal meta-analysis was not feasible.

⇒ The weighted z-test was used to quantify the strength of evidence and synthesise the findings, but only selected air pollutants were assessed due to lack of studies on other categories.

In low-income and middle-income countries (LMICs), NCDs are a major barrier to development.[1] Type 2 diabetes (T2D) is a large contributor to the NCD burden and a major risk factor for cardiovascular disease, affecting the quality of life and reducing life expectancy by about 4–8 years compared with those without the disease.[2] T2D is characterised by insufficient insulin production and inefficient insulin utilisation by the body[3] and reduction in β cell function.[4] Approximately 537 million people worldwide are affected by T2D and the number is predicted to rise to 783 million by 2045.[5 6] LMICs are expected to see an 87% increase in T2D prevalence, with the Indian subcontinent predicted to experience the largest increase.[7] More than 60% of people with T2D in the world are Asians.[8]

T2D is primarily explained by genetic, demographic and lifestyle factors. Recent findings, however, indicate that an increasing proportion of T2D is attributable to environmental factors including the built environment.[1 9] The built environment encompasses the physical and built context in which people live, work and socialise, including man-made

## INTRODUCTION

Non-communicable diseases (NCDs) are a key cause of poor health globally, and account for 7 out of 10 deaths worldwide.

structures such as buildings and transportation systems, and natural features such as parks and urban forests.[10 11] The built environment can have both positive and negative impacts on T2D. Factors such as ambient air pollution or a transport network which discourages active transport can have negative impacts on health.[12 13] Key built environment characteristics which improve public health include environments which promote physical activity, healthy eating and low pollution. The growth of cities without planned infrastructure to facilitate healthy lifestyle choices constrains opportunities for improved public health.[14] With healthier environments, 23% of deaths could be prevented worldwide.[15]

Approximately 4.7 billion people live in Asia, representing 60% of the world's population.[16] Yet, most of the empirical evidence on the built environment and T2D comes from Western countries which have a very different urban fabric compared with Asian cities. Asian cities are characterised by rapid, often unstructured, urban growth, high population density and, in some cases, temporal mix of land use.[17] To better understand the relationship between the built environment and T2D in Asia, it is, therefore, important to synthesise the available evidence in the Asian context, to identify gaps in knowledge and support local city planning and public health interventions.

## METHODS AND ANALYSIS
### Search strategy
A comprehensive systematic bibliographic search was conducted to identify epidemiological evidence on the associations of built environment characteristics with T2D in Asian countries. The systematic review followed the Preferred Reporting Items for Systematic Reviews and Meta-Analyses guidelines.[18] The protocol was registered with the International Prospective Register of Systematic Reviews (PROSPERO) database (CRD42020214852). A comprehensive search strategy identified peer-reviewed journal articles from inception to 23 January 2023 using the electronic bibliographic databases Medline, Embase and Global Health. The search strategy included combinations of keywords related to the built environment (urban green space, walkability, food environment, accessibility of services, air pollution, density and clustering measures), health outcome (T2D), study design (observational studies including cohort, cross-sectional and case–control studies) and study area (Asia). A reference search was carried out against newly identified articles until no more relevant articles were discovered.

### Patient and public involvement
There was no direct involvement of patients or the public in this study.

### Eligibility criteria
Study titles and abstracts were examined to assess their eligibility for inclusion based on predefined participant,

intervention, comparison and outcome criteria, shown in table 1. Only studies published in the English language were considered. As T2D accounts for the majority of diabetes cases in the general community (>90 %),[3] studies that did not specify the type of diabetes were included.

Access and availability of urban green space and higher walkability, which encourages people to walk in their local community, are both hypothesised to increase physical activity and consequently reduce T2D prevalence. The food environment facilitates either healthy or unhealthy food acquisition and consumption within the wider food system and can, therefore, have both positive and negative effects on T2D. Accessibility and availability of services including recreational facilities and healthcare services are hypothesised to decrease T2D prevalence. Long-term air pollution linked to local sources including traffic has been hypothesised to increase T2D prevalence.

### Screening, data extraction and preparation
Online supplemental file 1 provides a comprehensive overview of the search terms, including MESH terms used for MEDLINE, Embase and Global Health using OVID databases and extracted data using Microsoft Excel. Duplicates were removed and studies screened against the predefined study selection criteria, independently by two reviewers (AR and TSM), using Covidence, an online, systematic review tool. At all screening stages, title, abstract and full text, discrepancies were resolved through discussion with a mediator (RH). A standardised data extraction form was used for collection of information on methodology, outcome and exposure measures, including study characteristics (year of publication, study design, sample size, country), participant characteristics (gender, age range), health outcomes (T2D outcome and measurement method), built environment characteristics (type, measurement tool) and statistical methods. Data extraction was performed by reviewer TSM and data extraction was assessed by reviewer AR.

### Quality assessment
To assess the quality of included studies, the Newcastle-Ottawa Quality Assessment Scale (NOS) for Observational Cohort in an adapted version for cross-sectional studies[19] was used, following previous examples from the built environment literature.[20–22] The quality assessment was performed by reviewers AR and TSM, conflicts were discussed with reviewer RH and resolved. Each study was scored for quality by blinding the authors, institutions, country and journal. In evaluating the quality of each study, six criteria were used as follows: (1) representativeness of the sample, (2) study size, (3) ascertainment of the exposure, (4) comparable subjects in different outcome groups, (5) assessment of the outcome and (6) statistical test. Based on these criteria, star ratings were awarded. The six criteria were categorised into three broader categories relating to: (A) the study population, (B) the comparability of population and (C) characterisation of exposure/outcome variables.[23] The maximum number of

**Table 1** Inclusion and exclusion criteria for literature search

| | Inclusion criteria | Exclusion criteria |
|---|---|---|
| Population | Adults aged 18 years and above residing in Asia. Studies conducted on adults including specific populations such as pregnant women were included. | Animal studies and those involving specific populations such as pregnant women. |
| Intervention | Studies exploring any of the following built environment characteristics:<br>▶ Urban green space, including parks, ground cover vegetation, street trees, green roofs.<br>▶ Walkability, including land use mix, residential density, street connectivity.<br>▶ Food environment, including distance and density of health and unhealthy food outlets.<br>▶ Availability and accessibility to services, including distance and density of shops, healthcare and recreational facilities.<br>▶ Environmental pollutants: noise, air pollution. | Those that do not meet the criteria of built environment features and studies in an experimental setting. |
| Comparators | NA | NA |
| Outcome | Studies reporting outcomes on: prevalence/incidence of T2D and pre-diabetes, fasting blood glucose and 2-hour plasma glucose, glycated haemoglobin, insulin and Homeostatic Model Assessment for Insulin Resistance levels. | Studies that evaluated admissions to hospital or emergency departments due to T2D. |
| Study type | Observational studies on the association of built environment features with T2D outcomes: including cohort, cross-sectional and case–control studies. | Controlled trials, reviews, case reports and intervention studies. Specific publications which do not report original scientific research including letters, editorials, interviews and legal documents. |

NA, not available; T2D, type 2 diabetes.

stars awarded for the study population was five stars while the comparability of populations and characterisation of exposure/outcome variables were awarded one star and three stars, respectively. The study quality was evaluated based on the total number of stars assigned to each study, with 8–9 stars defined as 'very good' quality, 6–7 stars defined as 'good' quality, 4–5 stars defined as 'satisfactory' quality and 0–4 stars defined as 'unsatisfactory' quality. Online supplemental file 2 provides the summary of the quality assessment using the NOS.

### Data synthesis

Due to the heterogeneity in built environment characteristics, outcome measures and methodology used in the eligible studies, a formal meta-analysis was not feasible. Heterogeneity in built environment studies is a common issue that has been described by other authors.[20 22 24] To overcome this issue, a semiquantitative meta-analytical method was used, which has been developed and described in detail previously.[25 26] This method synthesises studies based on a weighted $z$-test, which considers the direction of findings, either in the expected or unexpected direction, and weights their level of significance based on the p value[27] as shown in table 2.

In order to calculate the z-score associated with a specific exposure–outcome relationship, the sum of z-values was divided by the square root of the sum of squared weights, using the quality assessment scores of individual studies as weights. After calculating the weighted z-value, the two-tailed p value was used to determine whether the evidence was strong or weak using the following criteria: p<0.05 for weak evidence, p<0.01 for strong evidence and p<0.001 for very strong evidence. The following equation was used to derive a summarised weighted z-value:

$$Weighted z = \frac{\sum weight_j z_j}{\sqrt{\sum weight_j^2}}$$

### RESULTS

Initially, 5208 articles were identified, of which 2115 were duplicates. After screening of articles according to eligibility criteria, 28 articles were eligible for inclusion in the systematic review (figure 1).

**Table 2** Significance levels and associated z-values used in semiquantitative meta-analysis

| Statistically significant | α | z-value |
|---|---|---|
| Yes (expected direction) | 0.05 | 1.96 |
| | 0.1 | 1.64 |
| No (null) | 0 | 0 |
| Yes (unexpected direction) | 0.1 | −1.64 |
| | 0.05 | −1.96 |

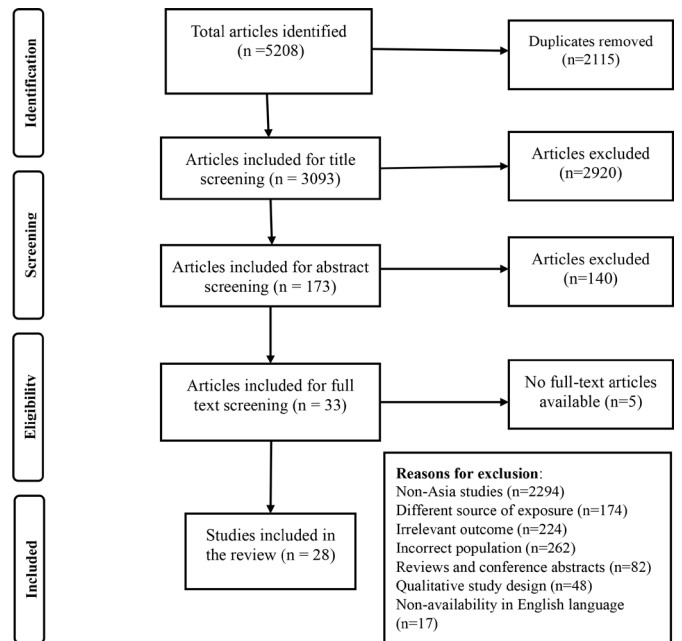

**Figure 1** Study identification, screening and eligibility, guided by PRISMA for T2D outcome. PRISMA, Preferred Reporting Items for Systematic Reviews and Meta-Analyses; T2D, type 2 diabetes.

The majority of studies were conducted in China (53.6%), followed by India (17.8%), South Korea (7.1%), Japan (3.6%), Taiwan (3.6%), Bangladesh (3.6%) and South Asia (Bangladesh and Sri Lanka) (3.6%). The earliest study was published in 2011, over 75% of studies after 2017. Cross-sectional study designs were used in 71% of the studies, while cohort and case-control study designs were used in 25% and 4% of the studies, respectively. All studies included both male and female participants. Sample sizes ranged from 120 to 3 41 211 participants, and different methods for participant recruitment and data collection were used, such as medical records and self-reported health status. Study sample sizes <1000 were used by 7%, and studies using sample sizes 1001–2500 and >2500 were used by 11% and 82%, respectively. The quality score ranged from 5 to 9. The mean quality score was 8.3 (SDs 0.85) indicating overall good and very good quality. The detailed characteristics from all the studies included in the systematic review are presented in online supplemental file 3.

The majority of studies used objectively collected data on diabetes (n=21), with the outcome being evaluated by biomedical tests or retrieved from registry records. One study used self-reported data[28] and three studies used a combination of hospital records and self-reported data.[29–31] The outcomes included diabetes and pre-diabetes prevalence and incidence (n=19), levels of fasting blood glucose (FBG) (n=11), 2-hour plasma glucose (n=1), glycated haemoglobin (HbA1c) (n=5), insulin level (n=1) and Homeostatic Model Assessment for Insulin Resistance (HOMA-IR) (n=1). Outcome variables were expressed as changes in continuous measurements

in longitudinal or cohort studies (eg, changes in FBG levels from baseline to follow-up) or as T2D prevalence in cross-sectional studies.

Studies explored a wide range of built environment characteristics on (1) urban green space (n=3), (2) walkability (n=2), (3) food environment (n=3), (4) availability and accessibility to services (n=2) and (5) air pollution (n=19). Studies used either objectively measured exposure derived using geographical information systems (GIS) (n=15) or audit measures (n=13). The sphere of influence of exposures was defined as either a circular buffer (n=3) or street-network buffer (n=1) around the participants' place of residence, ranging from 400 m to 1600 m. Some studies also used proximity to destinations (n=2) and modelled air pollution concentration at place of residences (n=8).

## Built environment and T2D

The results are presented for each built environment category separately. Study characteristics and quality assessment for all included studies are presented in online supplemental files 2 and 3, respectively. Due to fewer than five studies on most built environment categories, the weighted z-score was calculated only for air pollution.

## Urban green space

Three studies investigated the association of urban green space with T2D,[32–34] two studies conducted in China[33 34] and one in Taiwan.[32] All studies used FBG as a measure of T2D; one study used 2-hour glucose, HOMA-IR.[33] Greenness was assessed with Normalised Difference Vegetative Index (NDVI)[32 33] or Soil-Adjusted Vegetation Index,[33] one study assessed green space ratio, green vision index and evergreen tree configuration.[34] The statistical analysis was performed using logistic regression,[32] binary logistic regression[33] and linear mixed effect models.[32 34] Greenness was associated with lower glucose levels in Taiwan[32] and China[33] (p<0.05). Physical activity,[32] body mass index (BMI) and air pollution[33] were considered as potential mediators for this association. However, the other exposure variables used to measure urban green space did not show an association with T2D.[34]

## Walkability

Two studies assessed walkability using a cross-sectional design in Japan[35] and South Korea.[36] Both studies assessed T2D using FBG levels, one study assessed HbA1c levels and medical history, with study participants recruited from secondary data.[35] The study conducted in Japan assessed walkability using the availability of parks, slope and land use mix, while the study from South Korea used street patterns, slopes and neighbourhood amenities, such as the amount of neighbourhood park space, number of shopping malls and distance between communities and malls, which were both quantified and qualitatively assessed through site observations and GIS analyses. Multilevel regression[35] and multiple logistic regression[36] were used to perform the statistical analysis. According to

the study conducted in South Korea, people who lived in communities with better street networks, defined as grid layout of pedestrian pathways and pleasant landscape, including street trees, had a lower risk of T2D (OR 0.83, 95% CIs 0.77 to 0.91). Other associations of exposure variables with T2D were not statistically significant. The study from Japan did not find a statistically significant association of a hilly environment on T2D. Steeper slopes, on the other hand, reduced the risk of poorly controlled T2D (HbA1c levels of ≥7.5% in healthy people without any comorbidity or ≥8% in the elderly).

## Food environment

The food environment was examined by three studies.[35 37 38] Among participants aged 65 and older, one study examined the perception of access to grocery stores for fresh fruit and vegetables in Japan.[35] The South Asian Biobank study which analysed data from Bangladesh and Sri Lanka, collected information on fast food restaurants (FFR), supermarkets, mobile carts and stationary carts within 300 m buffer around participants' place of residence.[38] The study, conducted in India, collected information on shops and services selling food, tobacco and alcohol, as well as the density of vendors. All studies assessed T2D based on FBG levels, while two studies additionally assessed T2D with HbA1c and insulin levels (35.37) and the South Asian Biobank study additionally used high glucose levels and diagnosed diabetes. The study from Japan found no significant association between perception of access to grocery stores and T2D. Using three levels of mixed-effect linear regression, the density and distance of fruit and vegetable vendors and highly processed/takeout food vendors were analysed in relation to T2D in the Indian study. The South Asian Biobank study used ordinary least squares multivariate regression analysis to examine in association with T2D among adults and its heterogeneity according to income and gender. The associations between highly processed and take-away food vendor density and FBG (OR 0.30, 95% CI −0.14 to 0.74) and insulin (OR 0.13, 95% CI −0.04 to 0.29) were statistically non-significant.[37] There was no association between distance to the nearest fruit and vegetable vendor or highly processed and take-away food vendor and FBG or insulin levels.[35] An increase in blood glucose levels and the likelihood of being diagnosed with diabetes was associated with having at least one FFR locally by 16% (OR 1.6, 95% CI 1.01 to 1.33) and 19% (OR 1.19, 95% CI 1.03 to 1.38), respectively. FFR density had a stronger positive association with blood glucose levels among women than men. In contrast, FFR proximity had a stronger association with blood glucose levels among men and those with higher incomes.[38]

## Availability and accessibility to services

Two studies conducted in China[28] and Bangladesh[39] examined the relationship between the availability and accessibility of services such as healthcare, recreational facilities and T2D. Both studies used a cross-sectional study design and collected data from medical records. T2D was measured using FBG and self-reported T2D. Generalised linear mixed models[39] and ordinary least squares models[28] were used for statistical analyses. T2D was more prevalent in areas of greater distance from health facilities, such as community clinics, but the difference was statistically not significant (OR 1.05, 95% CI 0.6 to 1.61).[28] In the study conducted in Bangladesh, the association of distance to services with T2D was statistically non-significant except for the distance to a cinema hall (OR 1.76, 95% CI 1.04 to 2.98).

## Air pollution

Nineteen studies assessed air pollution in relation to T2D. Particulate matter (PM) of various particle sizes, black carbon (BC), nitrogen dioxide ($NO_2$), ozone ($O_3$) and sulfur dioxide ($SO_2$) were the main air pollutants studied. Eleven studies were cross-sectional, followed by cohort studies (n=7) and one case–control studies. There were 12 studies conducted in China, 3 in India and 1 each in South Korea, Iran, Taiwan and Malaysia. A total of 482 833 people were enrolled in a cross-sectional study with the largest sample size.[32] $PM_{2.5}$ (n=12)[29 31 40–48] was the most commonly assessed air pollutant in relation to T2D, followed by $PM_{10}$ (n=8),[30 41 49–54] $NO_2$ (n=6),[41 51–55] $SO_2$ (n=6),[49–54] $NO_x$ (n=2),[49 54] $O_3$ (n=2),[41 54] $PM_1$ (n=2)[44 55] and BC (n=1).[29] Six studies used medical history and T2D treatments to assess T2D status, while other studies used FBG, HbA1c and fasting plasma glucose levels, and one study used random capillary blood glucose to assess T2D. Several studies used secondary data such as data from Korea Community Health Survey data, Malaysian National Health, China Health and Retirement Longitudinal survey and Morbidity Surveys for health. Air pollution exposure was assessed using either measurements from local environmental monitoring centres and gravity samples, or air pollution models such as satellite-based spatial statistical models, spatiotemporal models and dispersion models.

Six studies reported that there was a significant positive association between air pollution and T2D among females but not males.[30 43 46 50 51 53] Two studies performed gender-specific stratified subgroup analyses and found a positive association for females but not for males.[46 50] In stratified analyses, exposure to air pollution has been associated with an increase in diabetes incidence among under 65 years compared with the elderly,[30 43 53] rural subjects in comparison to urban subjects,[43] BMI<25 kg/m² in comparison to BMI≥25 kg/m²,[53] non-smokers in comparison to smokers[43] and normotensive in comparison to hypertensive.[30 43] There is also evidence that air pollution has positive associations with T2D among the elderly,[51] overweight and obese individuals.[30]

Twelve studies evaluated $PM_{2.5}$ with 11 reporting a positive association between air pollution and T2D and 1 study reported a negative association.[39] Most studies indicate that the risk of T2D increased when exposure to $PM_{10}$ (n=8) or $SO_2$ (n=6) increased. One study reported no

association between $PM_{10}$ and T2D in men.[55] Six studies on $NO_2$ found a significant increase in T2D risk when exposed to higher levels of $NO_2$.

Using the z-test, we synthesised the evidence for $PM_{2.5}$, $PM_{10}$, $NO_2$ and $SO_2$, separately. The two-tailed p value from the weighted z-value for $PM_{2.5}$, $PM_{10}$, $NO_2$ and $SO_2$ and T2D was p<0.001, indicating a very strong association. The number of studies for $NO_x$, $O_3$, $PM_1$ and BC was too small to derive z statistics.

## DISCUSSION

This is the first systematic review to synthesise the evidence of the association of built environmental characteristics including urban green space, walkability, food environments, availability and accessibility to services (recreational and healthcare), and air pollution with T2D in Asia. We found strong evidence of an association of air pollutants such as $PM_{2.5}$, $PM_{10}$, $NO_2$ and $SO_2$ with higher incidence/risk/prevalence of T2D. Other built environment categories lacked consistent evidence to establish an association with T2D.

We found that measures for urban green space including green space ratio, green vision index and evergreen configuration were not significantly associated with T2D.[28 34 39] In contrast to our findings, some cross-sectional studies in Western countries found lower rates of T2D among people exposed to more green spaces, including studies from the Netherlands[56] and the UK.[57] A Norwegian study, however, did not find an association between green space and T2D, which the authors argued might be due to the fact that most of the residents had easy access to green spaces.[58] Similarly to the results of two studies conducted in Taiwan[32] and China,[33] a systematic review conducted globally to examine the association between greenness as measured by NDVI and T2D supported the notion that people and communities exposed to greenness, especially in their neighbourhood, have a lower risk of T2D.[59] South Asia's urban green space has been steadily diminishing. Green spaces and conservation areas are currently under threat, primarily because of unplanned and disorganised housing construction in cities.[60]

Walkability is hypothesised to increase physical activity, and therefore, might contribute to lower T2D rates. An effective neighbourhood design considers how land use decisions affect people's access to essential services in their communities. Completing, connecting and compacting a neighbourhood has been associated with health benefits such as increased physical activity, safer and easier transportation, enhanced employment productivity, and social inclusion. Strategies to encourage connectivity include enhancing walkability, establishing mixed land use, and designing complete and compact neighbourhoods.[61] An Australian study indicated that hilly neighbourhoods may prevent T2D,[62] while the study, conducted in Japan in a hilly neighbourhood, showed no link between walkability and T2D.[35] This difference might be due to the specific population group, which focused on the elderly in the Japan study. Higher built-up density and reduced greenness were found to be connected with lower T2D rates and physical inactivity in Kerala, India.[63] A study in South Korea found that participants in communities that have better street networks such as grid pattern of pedestrian sidewalks and attractive scenery were less likely to have T2D.[36] This is in line with findings from an Australian study indicating that study participants who lived in areas with better walkability were less likely to develop T2D.[64] Our systematic review found that there is currently not much research carried out on particular built environmental attributes related to accessibility to healthcare centres in Asia. The need for longitudinal studies is thus imperative to identify such specific built environment attributes that may influence T2D outcomes in order to establish future urban design regulations for healthier and more efficient communities. The effects of walking in neighbourhoods can be increased by providing pedestrian paths that are usable and unobstructed, by implementing motor-traffic reduction strategies, by increasing good street connectivity and by providing parks, green space, playgrounds and recreation areas.[65]

Our systematic review found consistent evidence of an association of the food environment with T2D risk/prevalence. An increase in fruit and vegetable vendor availability was associated with a lower T2D risk in an Indian study.[37] In this study, there was a strong evidence for an association between T2D risk and food vendor density within 400 m of the place of residence, while the association between take-away/processed food vendor availability and T2D was not significant. Two studies conducted in America and the UK discovered a link between increased number of fast-food restaurants and convenience stores and a higher prevalence of T2D.[66 67] Four studies conducted in the USA and Australia found no significant association between supermarket/grocery store availability and T2D prevalence.[67–71] Various sociocultural and economic norms influence how people purchase and consume food, which may explain differences in the association between food availability and T2D risk in different countries.[72]

We found a consistent and strong association between air pollution and T2D. Higher levels of $PM_{10}$ and $PM_{2.5}$ were associated with an increased risk of T2D incidence and prevalence; higher $NO_2$ exposure was associated with a higher T2D prevalence, but no statistically significant association with $O_3$ was found. In a previous systematic review on air pollution and T2D, with most studies from North America and Europe,[73] results were consistent with our study. Another meta-analysis showed consistent evidence of high exposure to air pollution being linked with greater T2D risk/prevalence[74] but identified a lack of studies from LMICs.

According to previous studies conducted in Western countries,[75–78] there is a stronger evidence of associations between air pollution and T2D among women compared with men. Clougherty suggested that sex-linked biological

differences in hormonal complement, body size, activity patterns or coexposures may explain such differences.[79] Women have been found to suffer more severe health effects due to air pollution, particularly among the elderly or when assessing residential exposure.[79] There is further evidence that women are more susceptible to air pollution due to hyper-responsive airways,[80] lung size, airway diameter[81] and residential environmental factors.[82] A Danish Nurse Cohort study reported that $PM_{2.5}$ and diabetes incidence increase in never-smokers and are more prominent in obese individuals.[76]

In this systematic review, we identified a paucity of evidence from a majority of countries in Asia. In particular, studies from Southeast Asia, including Indonesia, the fourth most populous country in the world, are lacking. Only seven cohort studies were conducted using a nationally representative sample while the other studies selected samples from specific regions such as specific cities, towns or counties. Hence, this systematic review might not be representative of the Asian population. Most of the studies in our review measured built environment attributes at a single point in time, however, longitudinal exposure data can be beneficial to detect changes in exposure in relation to changes is T2D.[83]

Overall, evidence from LMICs in Asia on the relationship between the built environment and T2D is still lacking. Despite the vast amount of literature exploring the traditional risk factors for NCDs, little attention has been directed towards specific factors of the built environment related to T2D outcomes in Asian populations. Depending on the quality and characteristics of the built environment, there can be large differences between developed and developing countries, which can affect T2D patterns among Asian residents.[84] The urbanisation trend in Asia is expected to undergo a significant shift in the next decade. In light of this circumstance, urban planners have a crucial opportunity to design smart cities and environments that encourage healthy lifestyles among their residents.[85] Nevertheless, this prime target cannot be achieved without a profound evidence base of the relationship between built environment and health and well-being.[86] Understanding the role that the built environment plays in influencing disease risk factors may help prevent the onset or deterioration of T2D with consequently vast benefits for public health.

## CONCLUSION

Our systematic review has identified several built environment characteristics, which were significantly related to T2D in Asia, in particular air pollution. Results also highlight potential effect modification by age and sex. Future studies should assess the modifying role of socioeconomic status and ethnicity. Such studies should be carefully designed to better understand the role of potential confounders, risk factors and effect modifiers. This particularly applies to LMIC where India and China already have a huge burden from T2D. Such evidence is

essential for public health and planning policies to (re) design neighbourhoods and help improve public health across Asian countries.

**Author affiliations**
[1]Department of Research Operations, Madras Diabetes Research Foundation, Chennai, Tamil Nadu, India
[2]School of Public Health, SRM Institute of Science and Technology, Kattankulathur, Tamil Nadu, India
[3]School of Public Health, Imperial College London, London, UK
[4]Centre for Health Economics and Policy Innovations, Imperial College Business School, London, UK
[5]Department of Translational Research, Madras Diabetes Research Foundation, Chennai, Tamil Nadu, India
[6]Department of Diabetology, Madras Diabetes Research Foundation, Chennai, Tamil Nadu, India
[7]MRC Centre for Environment and Health, School of Public Health, Imperial College London, London, UK

**Contributors** A search strategy was developed and guided by AR, TSM, DK, SAM and DF, while independent searches were conducted by AR and TSM, and conflicts were resolved by RH. The data were interpreted by SAM, DK and RP. After data synthesis and analysis, a draft of the manuscript was prepared by AR, shared with PV, VM, RMA and DF for their input. Corrections were made based on their suggestions. The accuracy and integrity of the work are investigated and resolved by RMA and DF. Corresponding authors DF and RMA corrected and approved the final version for submission. AR is responsible for the overall content as guarantor.

**Funding** This work was supported by the MRC Centre for Environment and Health, which is currently funded by the Medical Research Council (MR/S019669/1, 2019-2024). This study is part funded by the National Institute for Health Research (NIHR) Health Protection Research Unit in Chemical and Radiation Threats and Hazards, a partnership between UK Health Security Agency and Imperial College London (ICL). Infrastructure support for the Department of Epidemiology and Biostatistics was provided by the NIHR Imperial Biomedical Research Centre. The first author (GRA) worked with the ICL team as a visiting researcher in the School of Public Health as part of a training programme funded by the National Institute for Health and Care Research (NIHR's) Short Placement Award for Research Collaboration(NIHR GHR SPARC PILOT 01-19-06). The second author (TSM) was a student at ICL who pursued her M.Sc Epidemiology in the School of Public Health, entirely funded by the National Institute for Health and Care Research-Global Health Research Unit (NIHR GHRU) project. This research was also supported by the National Institute for Health Research (NIHR) (16/136/68) using UK aid from the UK Government to support global health research.

**Disclaimer** The views expressed in this publication are those of the authors and not necessarily those of the NIHR, UK Health Security Agency or the UK Department of Health and Social Care.

**Competing interests** None declared.

**Patient and public involvement** Patients and/or the public were not involved in the design, or conduct, or reporting, or dissemination plans of this research.

**Patient consent for publication** Not applicable.

**Provenance and peer review** Not commissioned; externally peer reviewed.

**Data availability statement** All data relevant to the study are included in the article or uploaded as online supplemental information.

and indication of whether changes were made. See: https://creativecommons.org/licenses/by/4.0/.

**ORCID iDs**
Garudam Raveendiran Aarthi http://orcid.org/0000-0002-7742-5301
Rajendra Paradeepa http://orcid.org/0000-0002-4909-3733

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
