## [Reviewer comments · BMJ Open]

ARTICLE DETAILS

TITLE (PROVISIONAL)	Associations of the built environment with Type 2 diabetes in Asia: A systematic review
AUTHORS	Raveendiran, Aarthi; Mehreen, T S; Moosawi, Suzana; Kusuma, Dian; Harish, Ranjani; Pradeepa, Rajendra; Venkatasubramanian, Padma; Mohan, V; Anjana, Ranjit Mohan; Fecht, D

VERSION 1 – REVIEW

REVIEWER	Martins Pereira , Gonçalo Instituto Português de Oncologia de Lisboa Francisco Gentil EPE
REVIEW RETURNED	01-Oct-2022

GENERAL COMMENTS	In this manuscript, the authors present a systematic review on the impact of built environment characteristics on type 2 diabetes risk in Asia. First, I would like to congratulate the authors on this interesting topic and review. Nowadays, there is a growing awareness and focus on the environmental risk factor for type 2 diabetes due to its continuous increasing incidence. Since Asia is expected to become the second IDF region with the greatest number of people with diabetes by 2045 this review brings attention not only to known and potential environmental risk factors for type 2 diabetes but also for their impact in this world region, so that mitigation strategies may be applied. I believe the authors point to the limitations found in their review, highlighting the need for further studies. There are a few important aspects in the manuscript that I believe need major revision: 1 - I am not entirely sure whether the inclusion of the term "meta-analysis" in the title is appropriate because it was not done due to lack of studies.2 - The abstract must be thoroughly reviewed as it has some spelling errors and does not mention how many studies were included in the systematic review.3 - Throughout the text some typos and spelling errors were identified.4 - In line 115, the authors mention that "(...) diabetes accounts for the majority of T2D cases (...)" something that is incorrect because it should be the other way around, that is, "(...) T2D accounts for the majority of diabetes cases (...)".5 - In the results section, the authors mention that "Sample sizes ranged from 86 to 7,73,602 participants (...)". However, when consulting the online supplement file 3, it appears that these numbers of participants do not correspond to the studies included. Also, and perhaps more worrisome, some numbers of participants are incorrect after consulting the original articles. Finally, the
--

	reference's numbers in the online supplement file 3 do not match the bibliography numbering. 6 - In the conclusions, the authors refer to "Our systematic review has identified significant gender differences (...)". However, in the results the only reference to gender is on line 292 and this fact is not discussed in the discussion section. Thus, I think that stating "significant gender differences" as one of the main conclusions of the systematic review is perhaps a bit too much. 7 - Despite not knowing the referencing style used by the journal, some references are incorrectly referenced, regardless of the referencing style used. Overall, I believe this manuscript brings an important contribution to this field, therefore I advise to accept this manuscript for publication, pending the aforementioned major revisions.
--	---

REVIEWER	Lin, Chien-Yu Waseda University
REVIEW RETURNED	25-Dec-2022

GENERAL COMMENTS	Thank you for the opportunity to review this review paper that addresses an important public health issue. The procedures of this systematic review, which follow the PRISMA guidelines, are methodical by and large. However, I have several comments for the authors and hope these comments can be helpful for them to improve this paper to be further organised and acceptable to be published. 1. My primary concern is that this paper needs a better justification to explain the reason for reviewing the associations between built environment characteristics and diabetes only for studies from Asian countries. The authors argued that research focused on the context of non-Asian settings, but studies on Asian cities are lacking to assist in future city planning and public health strategies; this statement may be more proper for the original paper using an Asian sample but not valid for a review paper. If near-zero or few studies have investigated this topic, only few studies can be systematically searched, and the insufficient findings are hard to synthesise. The authors need to emphasise the importance and need of summarising these findings from Asian countries. 2. This paper can be largely improved if the authors provide a clear and specific framework for categorising the built environment characteristics studied. (1) First, the categories of built environment characteristics listed in the Methods section are suggested to be in accordance with the subtitles in the Results section (i.e., urban green space, walkability, food environment, availability and accessibility to services, and air pollution). (2) Second, it seems that the authors also reported the results and discussed them stratified by measure (e.g., space and greenness for the urban green space). However, such information needs to be well-organised. Developing a framework for all the built environment characteristics identified for this review paper may be helpful. (3) Third, previous studies investigating the associations of built environments with health outcomes usually include aesthetics and crime/traffic safety. The authors also include "aesthetic*" (#22) among the search terms. I wonder there is any reason or consideration for not including the characteristics as one category of built environment characteristics.
---

	(4) Fourth, the definition and hypothesis of the associations between built environment characteristics and health outcomes are suggested to be clarified in the Methods section rather than in the Results section. 3. As the review paper includes observational studies only (the causal relationships are not inferred), I suggest not using the term “impact” while describing the aims or the findings throughout the paper. 4. In the Abstract, it is suggested to report the direction of the association of particulate matter, nitrogen dioxide, and sulfur dioxide with type 2 diabetes risk. 5. I suppose that the exclusion criteria of this review paper are those only examining specific populations (e.g., pregnant women); if a paper examining adults (including some pregnant women) would be included? Or both these circumstances would be excluded in this paper 6. While extracting the important information from these included studies, I suggest reporting the confounders considered in each study. Especially, whether these studies included important confounders such as age, sex, and socioeconomic status may somewhat impact the quality of the study and the credibility of the research findings. Furthermore, such information can echo the suggestions on identifying the potential modifiers the authors made for future studies. 7. I suggest that the quality assessment should be conducted by two independent evaluators. The agreement between two evaluators can be reported and confirm its reliability. 8. I am quite confused that the authors concluded that the review paper identified significant gender differences. I suppose this statement is according to six studies showing statistical evidence for negative associations of air pollutants with diabetes risk factors among women, the elderly, and vulnerable people (although I am not sure what the vulnerable here refers to). I suggest separating a paragraph to provide more information while reporting the findings for stratified analyses from the included studies. For example, it needs to be clarified how many studies examined differences across subgroups and of which how many studies did report differences. Also, did the authors conduct stratified analyses using z-tests?
--	---

REVIEWER	Adebusoye, Busola University of Nottingham
REVIEW RETURNED	30-Dec-2022

GENERAL COMMENTS	Thank you very much for the opportunity to review this article. Please find below my comments. 1. Could you please check the spelling of synthesise on line 38 and 64. It should be synthesise not syntheses 2. Line 105 – it would be nice to specify the date to which the references were updated.
--

	3. Line 126 – Could you please specify the tool that was used to develop the extraction form e.g Microsoft Excel etc 4. Line 177 – Three different study designs are included in the review. I am concerned if the quality scale that you have used suffices for the three different designs. For example, one of the questions asked if the subjects in different outcome groups are comparable, I wonder how you were able to assign a score for studies with cross-sectional study designs with just one group. You may consider using different critical appraisal checklist for different study designs e.g JBI or anyone that you are familiar with. Otherwise, provide sufficient justification for using the same quality checklist for different study designs. 5. There are instances across the manuscript where there are no spaces between words, please read through and correct these. E.g Line 211- greenareas, Line 225 -walkin
--	---

VERSION 1 – AUTHOR RESPONSE

Reviewer 1's Comment

R1.1. In this manuscript, the authors present a systematic review on the impact of built environment characteristics on type 2 diabetes risk in Asia. First, I would like to congratulate the authors on this interesting topic and review. Nowadays, there is a growing awareness and focus on the environmental risk factor for type 2 diabetes due to its continuous increasing incidence. Since Asia is expected to become the second IDF region with the greatest number of people with diabetes by 2045 this review brings attention not only to known and potential environmental risk factors for type 2 diabetes but also for their impact in this world region, so that mitigation strategies may be applied. I believe the authors point to the limitations found in their review, highlighting the need for further studies.

RR1.1. We thank the reviewer for these encouraging comments.

R1.2. I am not entirely sure whether the inclusion of the term “meta-analysis” in the title is appropriate because it was not done due to lack of studies.

RR1.2. We agree with the reviewer that the term “meta-analysis” could be misleading in the context of our systematic review. We have revised the title accordingly, also with regards to comments by the Editor (E3) to “Associations of the built environment with Type 2 diabetes in Asia: A systematic review”.

R1.3. The abstract must be thoroughly reviewed as it has some spelling errors and does not mention how many studies were included in the systematic review.

RR1.3. We have carefully reviewed the abstract and corrected any spelling and grammar errors and restructured the abstract according to the editor's instructions. We now have mentioned the number of included studies in the abstract:

“Out of 5,208 identified studies, 28 studies were included in this systematic review.”

R1.4. Throughout the text some typos and spelling errors were identified.

RR1.4. We have carefully reviewed the manuscript and corrected any spelling and grammar issues.

R1.5. In line 115, the authors mention that "(...) diabetes accounts for the majority of T2D cases (...)" something that is incorrect because it should be the other way around, that is, "(...) T2D accounts for the majority of diabetes cases (...)"

RR1.5. We thank the reviewer for spotting this error and have now corrected this in the text, as follows: "As T2D accounts for the majority of diabetes cases in the general community (>90 %)...".

R1.6. In the results section, the authors mention that "Sample sizes ranged from 86 to 7,73,602 participants (...)". However, when consulting the online supplement file 3, it appears that these numbers of participants do not correspond to the studies included. Also, and perhaps more worrisome, some numbers of participants are incorrect after consulting the original articles. Finally, the reference's numbers in the online supplement file 3 do not match the bibliography numbering.

RR1.6. We thank the reviewer for spotting these inconsistencies. We have carefully reviewed all sample sizes against the original articles and made corrections where necessary. "Sample sizes ranged from 120 to 341,211 participants, and different methods for participant recruitment and data collection were used such as medical records and self-reported health status." We have further checked all references throughout the manuscript, tables and supplement and made necessary changes to the bibliography numbering.

R1.7. In the conclusions, the authors refer to "Our systematic review has identified significant gender differences (...)". However, in the results the only reference to gender is on line 292 and this fact is not discussed in the discussion section. Thus, I think that stating "significant gender differences" as one of the main conclusions of the systematic review is perhaps a bit too much.

RR1.7. We have expanded the results section to highlight more studies that focus on gender differences and have identified six studies which have shown a stronger association of air pollution on T2D in women compared to men: "Six studies reported that there was a significant positive association between air pollution and T2D among females but not in males".

We also added a point to the discussion that provides some explanation as to why this might be: "According to previous studies conducted in Western countries (75-78), there is stronger evidence of associations between air pollution and T2D among women compared to men. Clougherty et al., suggests that sex-linked biological differences in hormonal complement, body size, activity patterns, or co-exposures may explain such differences (79). Women have been found to suffer more severe health effects due to air pollution, particularly among the elderly or when assessing residential exposure (79). There is further evidence that women are more susceptible to air pollution due to hyperresponsive airways (80), lung size, airway diameter (81) and residential environmental factors (82). A Danish Nurse cohort study reported that PM2.5 and diabetes incidence increase in never-smokers and are more prominent in obese individuals (76)."

We have also amended the conclusion to tone down the focus on gender differences:

"Our systematic review has identified several built environment characteristics which were significantly related to T2D in Asia, in particular air pollution. Results also highlight potential effect modification by age and sex. Future studies should assess the modifying role of socioeconomic status and ethnicity. Such studies should be carefully designed to better understand the role of potential confounders, risk factors and effect modifiers. This particularly applies to LMIC wherein India and China already have a huge burden from T2D. Such evidence is essential for public health planning policies to (re)design neighbourhoods and help improve public health across Asian countries."

R1.8. Despite not knowing the referencing style used by the journal, some references are incorrectly referenced, regardless of the referencing style used.

RR1.8. We have carefully reviewed the referencing style throughout and made corrections if necessary to adhere to the journal style.

Reviewer 2's Comment

R2.1. My primary concern is that this paper needs a better justification to explain the reason for reviewing the associations between built environment characteristics and diabetes only for studies from Asian countries. The authors argued that research focused on the context of non-Asian settings,

but studies on Asian cities are lacking to assist in future city planning and public health strategies; this statement may be more proper for the original paper using an Asian sample but not valid for a review paper. If near-zero or few studies have investigated this topic, only few studies can be systematically searched, and the insufficient findings are hard to synthesise. The authors need to emphasise the importance and need of summarising these findings from Asian countries.

RR2.1. We have now highlighted the rationale for focusing on Asia more clearly in the introduction: “Approximately 4.7 billion people live in Asia, representing 60% of the world’s population (16). Yet, most of the empirical evidence on the built environment and T2D comes from Western countries which have a very different urban fabric compared to Asian cities. Asian cities are characterised by rapid, often unstructured, urban growth, high population density and, in some cases, temporal mix of land use (17). To better understand the relationship between the built environment and T2D in Asia, it is, therefore, important to synthesise the available evidence in the Asian context, to identify gaps in knowledge and support local city planning and public health interventions.”

R2.2. This paper can be largely improved if the authors provide a clear and specific framework for categorising the built environment characteristics studied.

RR2.2. We thank the reviewer for these very helpful suggestions.

(1) First, the categories of built environment characteristics listed in the Methods section are suggested to be in accordance with the subtitles in the Results section (i.e., urban green space, walkability, food environment, availability and accessibility to services, and air pollution). We have carefully reviewed the built environment characteristics and made sure that the built environment characteristics listed in the Methods are in accordance with the subtitles in the Results section.

(2) Second, it seems that the authors also reported the results and discussed them stratified by measure (e.g., space and greenness for the urban green space). However, such information needs to be well-organised. Developing a framework for all the built environment characteristics identified for this review paper may be helpful.

We have also included a detailed definition of the built environment characteristics in table 1:

“Studies exploring any of the following built environment characteristics:

- Urban green space, including parks, ground cover vegetation, street trees, green roofs;
- Walkability, including land-use mix, residential density, street connectivity;
- Food environment, including distance and density of health and unhealthy food outlets;
- Availability and accessibility to services, including distance and density of shops, health care and recreational facilities;
- Environmental pollutants: noise, air pollution.”

(3) Third, previous studies investigating the associations of built environments with health outcomes usually include aesthetics and crime/traffic safety. The authors also include “aesthetic*” (#22) among the search terms. I wonder there is any reason or consideration for not including the characteristics as one category of built environment characteristics.

We have also added the hypothesised relationship with T2D in the methods to highlight the expected direction of associations:

“Access and availability of urban green space and higher walkability which encourages people to walk in their local community, are both hypothesised to increase physical activity and consequently reduce T2D prevalence. The food environment facilitates either healthy or unhealthy food acquisition and consumption within the wider food system and can, therefore, have both positive and negative effects on T2D. Accessibility and availability of services including recreational facilities and healthcare services are hypothesized to decrease T2D prevalence. Long-term air pollution linked to local sources including traffic has been hypothesised to increase T2D prevalence.”

(4) Fourth, the definition and hypothesis of the associations between built environment characteristics and health outcomes are suggested to be clarified in the Methods section rather than in the Results section.

We couldn't find studies related to aesthetics in Asia that passed our inclusion criteria.

R2.3. As the review paper includes observational studies only (the causal relationships are not inferred), I suggest not using the term "impact" while describing the aims or the findings throughout the paper.

RR2.3. According to the reviewer's suggestion, we do not use the term 'impact' anymore and have replaced this with 'association' to reflect the observational nature of included studies throughout the manuscript.

R2.4. In the Abstract, it is suggested to report the direction of the association of particulate matter, nitrogen dioxide, and sulfur dioxide with type 2 diabetes risk.

RR2.4. We have added this information to the abstract: "We found very strong evidence of a positive association of particulate matter (PM_{2.5}, PM₁₀), nitrogen dioxide and sulfur dioxide ($p < 0.001$) with T2D risk."

R2.5. I suppose that the exclusion criteria of this review paper are those only examining specific populations (e.g., pregnant women); if a paper examining adults (including some pregnant women) would be included? Or both these circumstances would be excluded in this paper

RR2.5. Thank you for the comment. We have now made this clearer in table 1: "Adults aged 18 years and above residing in Asia. Studies conducted on adults including specific population such as pregnant women were included."

R2.6. While extracting the important information from these included studies, I suggest reporting the confounders considered in each study. Especially, whether these studies included important confounders such as age, sex, and socioeconomic status may somewhat impact the quality of the study and the credibility of the research findings. Furthermore, such information can echo the suggestions on identifying the potential modifiers the authors made for future studies.

RR2.6. We have now included information on confounders in online supplement 3; lines: 293-294.

R2.7. I suggest that the quality assessment should be conducted by two independent evaluators. The agreement between two evaluators can be reported and confirm its reliability.

RR2.7. The quality assessment was performed by reviewers GRA and TSM, conflicts were discussed with reviewer HR and resolved.

R2.8. I am quite confused that the authors concluded that the review paper identified significant gender differences. I suppose this statement is according to six studies showing statistical evidence for negative associations of air pollutants with diabetes risk factors among women, the elderly, and vulnerable people (although I am not sure what the vulnerable here refers to). I suggest separating a paragraph to provide more information while reporting the findings for stratified analyses from the included studies. For example, it needs to be clarified how many studies examined differences across subgroups and of which how many studies did report differences. Also, did the authors conduct stratified analyses using z-tests?

RR2.8. We have expanded on sex differences in the text to bring out the differential effect of air pollution on T2D by sex and other factors more clearly in the text: "Six studies reported that there was a significant positive association of air pollution and T2D among females but not in males (30,42,46, 50,51, 53). Two studies performed gender-specific stratified subgroup analyses and found a positive association for females but not for males (46,50). In stratified analyses, exposure to air pollution has been associated with an increase in diabetes incidence among the under 65 years compared to the

elderly (30,43,53), rural subjects in comparison to urban subjects (43), BMI <25 in comparison to BMI ≥ 25 (53), non-smokers in comparison to smokers (43), and normotensive in comparison to hypertensive (30,43). There is also evidence that air pollution has positive associations with T2D among the elderly (51), overweight and obese (30).”

In computing the weighted z-test, we applied weights for results from stratified analyses.

Reviewer 3's Comment

R3.1. Could you please check the spelling of synthesise on line 38 and 64. It should be synthesise not syntheses

RR3.1. We thank the reviewer for spotting this error and have made those corrections in the text.

R3.2. Line 105 – it would be nice to specify the date to which the references were updated.

RR3.2. As suggested by the reviewer, we have added the date of the literature search as follows: “A comprehensive search strategy identified peer-reviewed journal articles from inception to 23rd January 2023 using the electronic bibliographic databases Medline, Embase and Global Health.”

R3.3. Line 126 – Could you please specify the tool that was used to develop the extraction form e.g Microsoft Excel etc

RR3.3. We have used Microsoft Excel for the data extraction and have added this information to the text as follows: “Online Supplement File 1 provides a comprehensive overview of the search terms, including MESH terms used for MEDLINE, Embase, and Global Health using OVID databases and extracted data using Microsoft Excel. Duplicates were removed and studies screened against the pre-defined study selection criteria, independently by two reviewers (GRA and TSM), using Covidence, an online, systematic review tool.”

R3.4. Line 177 – Three different study designs are included in the review. I am concerned if the quality scale that you have used suffices for the three different designs. For example, one of the questions asked if the subjects in different outcome groups are comparable, I wonder how you were able to assign a score for studies with cross-sectional study designs with just one group. You may consider using different critical appraisal checklist for different study designs e.g JBI or anyone that you are familiar with. Otherwise, provide sufficient justification for using the same quality checklist for different study designs.

RR3.4. The decision to use the same quality assessment scale for all study designs follows other systematic reviews on the built environment and health, for example Malacarne et al used modified Newcastle–Ottawa scale for quality assessment and den Braver et al used Quality assessment tool for quantitative studies.(21,22) We have added this justification to the text: “To assess the quality of included studies, the Newcastle - Ottawa Quality Assessment Scale (NOS) for Observational Cohort in an adapted version for cross-sectional studies (19) was used, following previous examples from the built environment literature (20-22).”

R3.5. There are instances across the manuscript where there are no spaces between words, please read through and correct these. E.g Line 211- greenareas, Line 225 -walkin

RR3.5. We thank the reviewer for spotting this formatting issue and have corrected this in the text, if necessary, after careful checking of the manuscript.